# How Can the University Environment Support Student Quality of Life? A Novel Conceptual Model

**Alyson Lamont Dodd** [1,*] , **Georgia Punton** [2] , **Joanna Mary Averill McLaren** [3] , **Elizabeth Sillence** [1] **and Nicola Byrom** [3]

1   Department of Psychology, Northumbria University, Newcastle upon Tyne NE1 8ST, UK
2   Department of Psychology, Durham University, Durham DH1 3HP, UK
3   Department of Psychology, Institute of Psychiatry, Psychology and Neuroscience, King's College London, London WC2R 2LS, UK
*   Correspondence: alyson.dodd@northumbria.ac.uk

**Abstract:** During emerging adulthood (18–25 years), university students have taken steps towards independent living and learning. However, they are also in a liminal phase before the 'stable roles' of adulthood. This developmental context distinguishes them from both adolescents and peers who are not attending university. In order to support student well-being, their unique priorities and concerns need to be taken into consideration. This qualitative study explored what life domains were important to students, and what influenced Quality of Life (QoL) within these, in order to build a novel conceptual model of student QoL. Individual interviews ($n$ = 18) were conducted with undergraduate students (aged 18–25 years). The themes derived via Template Analysis were Supportive and Rewarding University Studies, Personal Growth, Social Support, Concerns about Finances and Financial Independence, Physical Environment, Physical and Mental Well-being, and Maintaining Balance. As well as conceptualising QoL in students, this model has practical value for operationalising student QoL. It is a framework to help universities to understand the needs and priorities of students and provide well-being initiatives in line with these needs.

**Keywords:** quality of life; well-being; transitions to adulthood; higher education; university; qualitative

## 1. Introduction

There is increasing concern about student mental health and well-being in the UK and worldwide [1,2]. The University Mental Health Charter [2] emphasises supporting mental health and well-being across the university community in key domains: staff working conditions; teaching, learning, and transitions; physical and cultural environments; and university support, both formal support for students with mental health difficulties, and proactive interventions that support the well-being of all students.

While outcome measures for both mental health *and* well-being are central to assessing impact [2], we need to distinguish between mental health symptom measures for the sub-group of students experiencing clinically significant difficulties, and well-being measures for the whole student population [3]. However, clinical measures are often used to measure the latter [4]. When well-being measures are used, these were developed for use in the general population, and may not reflect student priorities [4]. These measures focus on feeling good and functioning well, but when students were asked how *they* think their well-being should be measured [5], they valued multiple outcomes that included, but went beyond, more classical hedonic and eudaemonic perspectives of well-being [6–8], such as coping, social support and work–life balance.

Quality of Life (QoL) captures objective factors beyond how someone is currently feeling and functioning. These can be targeted to produce improvements, providing a pathway for enhancing well-being. Existing QoL models are multidimensional and broadly

map on to the domains outlined in the University Mental Health Charter [2]: physical and mental health, education and personal development, social support and relationships, finances, the physical [9,10]. The specific developmental and environmental context of students is not captured by existing models, yet QoL should be framed around the values, concerns, and environmental and social/cultural context of a group [10]. The majority of students are emerging adults (aged 18–25 years), meaning that they have taken steps towards independence, unlike most adolescents. However, they are not settled into the 'stable roles' of adulthood (e.g., careers), and experience different life transitions compared to emerging adults who have done so [11].

Despite this, generic adult and adolescent measures [9,10,12–14] are used in students. This is a problem because students may fail to see the relevance to their experience, hampering design, evaluation and ultimately meaningful impact of university well-being initiatives. The aim of the present study was to build a conceptual model of QoL in the social and cultural context of being an undergraduate student in the UK.

## 2. Method

*Participants*

Undergraduate students (aged 18–25) were recruited from three universities of varying size, age, and location (North East England, South East England, and London). The study was advertised via email, virtual learning environments, social media, and on-campus posters. We aimed to recruit twenty participants, a sample small enough to capture individual perceptions while sufficient to identify patterns across a range of perspectives [15]. Eighteen students completed interviews (see Table 1 for participant characteristics).

**Table 1.** Participant characteristics.

| Variable | Categories | *n* |
|---|---|---|
| Gender | Male | 6 |
| | Female | 12 |
| Age | 18–19 | 8 |
| | 20–21 | 7 |
| | 22–25 | 3 |
| Year of study | 1 | 7 |
| | 2 | 2 |
| | 3+ | 9 |
| Student status | Home | 12 |
| | International | 6 |
| Ethnicity | White British | 10 |
| | Any other white background | 4 |
| | Mixed | 1 |
| | Black or Black British | 1 |
| | Asian or Asian British | 2 |

## 3. Materials and Methods

Approval was granted by the (PNM Research Ethics Panel, Kings College London). Before taking part in individual interviews, participants completed an online consent form and screening questionnaire asking for gender, ethnicity, student status (home (UK), EU and international), and year of study. At the start of the semi-structured interview, QoL was explained using existing definitions [9,10]. Initial questions asked about what was important to QoL in the context of being an undergraduate student, what determines 'a good life' and what challenges this. Participants were then shown a wheel of well-being as

a visual prompt illustrating potential QoL domains. Interviews were digitally recorded and transcribed verbatim. Participants were given a £10 voucher as a thank you.

*Analysis*

Template Analysis [16,17] was selected for its flexibility in both epistemology and approach. An initial template comprised *a priori* codes from the wheel for well-being shown to participants (Table 2). Guided by the template, the first four transcripts were coded by multiple researchers to capture different perspectives (including student peer researchers) for quality checking [17]. After discussion, *a priori* codes were modified or removed to reflect the data, and a second template was applied. This iterative process continued until we were satisfied that no further modifications were needed. At this stage, the template was sense-checked by a panel of ten students (four male) who met the same eligibility criteria as for interviews. Following this, codes were clustered into salient themes and subthemes.

**Table 2.** A priori codes and final themes from Template Analysis.

| A Priori Codes | Themes | Subthemes |
|---|---|---|
| Academic Studies | Supportive and Rewarding University Studies | Supportive University Environment<br>Valuing and Enjoying the Course<br>Academic Pressure |
| Personal Growth | Personal Growth | Transitions and Personal Development<br>Building Confidence for Employability |
| Friends and Family | Social Support | Friendships and Peer Relations<br>Family |
| Romance/significant other | -- | -- |
| Money | Concerns about Finances and Financial Independence | |
| Physical Environment | Physical Environment | Clean, Safe and Social Living Space<br>Campus and the Wider Setting |
| Health | Physical and Mental Well-being | Having Sufficient Mental Well-being to Engage<br>Being Healthy and Living Well |
| Fun and Recreation | Maintaining Balance | Time Management and Balancing Priorities<br>Opportunities to Take a Break |

## 4. Results

Seven themes represented dimensions of student QoL and positive and negative influences within these (see Table 2). Across themes, the developmental context of emerging adulthood and the specific challenges of the transition into and through university were central.

### 4.1. Supportive and Rewarding University Studies

This theme focused on academic aspects of student life, divided into three subthemes. The first, *Supportive University Environment*, captured accessible support from university systems and staff. This was especially important in the context of transitioning to independent living and learning.

> "Usually in that very alien environment when you first join it, I think at least in our generation it does affect people's [QoL]—I think being supported is absolutely essential, so our support office and [student] help desk and Nightline always being accessible." Ppt 14

> "There has been a lot of uproar in the news about lack of student support and that obviously has a massive impact on quality of life. And if you're away from home for the first time you're obviously entering the adult world and if you don't

have any interaction with any adults, if you go straight into halls and all the 18-year-olds are your natural support network it's not necessarily as it should be. University has a lot of responsibility which they often do not fulfil to support 18-year-olds who have just left home for the first time because their only other support for information is people in the exact same position as them." Ppt 2

This emphasises the distinction between university and peer support. Participants value 'adult' support and guidance, and feeling listened to on a human level, while they navigate independence.

"I'm quite friendly and comfortable with all the lecturers . . . And having the student reps, so I have some say in how the course is structured so it's all a lot more open rather than being told this is what you're going to do. You're the students, we're the lecturer, and this is the only space you're allowed to work in—it's a lot more open which I like a lot in comparison to the school environment." Ppt 17

However, participants were also uncertain as to how much support they could ask for as independent learners.

"I would like more guidance but I'm not supposed to ask for it because I'm supposed to be working independently. So it's a conflict about what to do . . . I feel like maybe just advertising themselves more, like the tutors, saying you can always ask for help." Ppt 15

The second subtheme, *Valuing and Enjoying the Course*, captured enjoyment and fulfilment in their studies, interesting course content and having a passion for learning.

"I feel like intrinsically I do need to like you know what I am learning, and I do really do love—what I'm learning." Ppt 1

Participants appreciated that they were studying something they had chosen and had control over how to approach studying, unlike in compulsory education.

"I'm making progress it's on my terms, I'm choosing to go there, am I leaving now am I leaving in half an hour am I going to go for a long night? I guess, that and me choosing to do it makes it easier because it's controlled by me." Ppt 11

The third subtheme, *Academic Pressure*, captured the ingredients that contributed to the pressure cooker of student life: workload, parental expectations, social comparison, and uncertainty and lack of confidence around independent study.

"I think that [independent study] is one of the really good parts of university but the transition is quite scary and assignments looming in the back of your head—like should I be doing this? Should I not? Or am I working too hard or am I not working hard enough?" Ppt 11

In addition to being a stressor, workload and independent study also contributed to a sense of feeling productive and fulfilled.

"Exam and course stresses can affect quality of life but then I think you need those for goals as well so a good balance of those. So if you have too many deadlines or too much pressure then it can affect your quality of life but I feel like you need them for goals because otherwise you're not really working towards anything." Ppt 2

"It feels like they give you work and push you out the door and say go and do it, but once you're used to it its actually quite helpful to learn how to do things independently but at the beginning you don't really know what is going on." Ppt 15

Participants also experienced pressure around their academic performance and potential employability, framed by parental expectations and comparison to course mates.

"My dad is very like 'you're here, you need to do the best that you can. I'm providing for you I have the best dream for you I've given you everything that I didn't have' . . . sort of politely saying don't mess this up so there is that sort of unofficial strain." Ppt 1

"I tend to compare myself more to [people on my course] and how well they're doing and succeeding in their projects because I feel like they're the people that I'm going to be up against in the real world and when I look at the grades they're getting and the projects they're doing and what they're coming out with—I find myself thinking that they're better than me." Ppt 9

### 4.2. Personal Growth

Personal growth beyond academic studies was also important for QoL. Participants were trying to make the best of their university years and transition to adulthood, while also having an eye on the future and their employability. The subtheme *Personal Development and Transitions* emphasised that, as emerging adults striking out on their own for the first time, university was a unique opportunity for personal development alongside gaining independence and learning to manage life responsibilities.

"This new independence that you have . . . suddenly I have to do my own washing and organise my own life. Comparing myself to the me that started like two, three years ago it is crazy like I am a different person . . . I feel like if I'd been at home and stayed in my environment and was just working back there, I wouldn't have had that opportunity to grow and live in a city and meet all these new people and I would probably be the same basically, I think it has been really important." Ppt 9

"University should not just consider people going to get a degree and that's all they're there for they should grow into a fully functioning adult . . . I feel like people who haven't been through university as an undergraduate wouldn't have gone through the same growth as if they'd just stayed at home and went for a job or something like that. I think there is a lot of independence to be gained that impacts your quality of life." Ppt 2

The emphasis in the subtheme *Building Confidence for Employability* was on maximising future career prospects by obtaining transferable skills and experience relevant to future employment alongside their degree.

"I'm paying to get the best education I can so that I can get the best degree I can so I can unlock the best choice I can to have the best option in my future . . . things like employability. How do I best develop my skills so that I can work?" Ppt 11

"I need to get like an internship and worrying if I don't do that like—it's all going to be a waste and then I'm not going to end up doing what I want to do and what is it exactly that I want to do?" Ppt 9

This uncertainty about what they wanted from a future career and what goal they were working towards impacted their current experience.

"I don't feel like I have an ultimate goal at the moment. I don't know what I want to do in the future, I have an inkling but I'm not that sure and I haven't really found something I'm that passionate about so it would be nice to just have a lightbulb moment and know what I want to do because that would give me some more purpose and fulfilment." Ppt 3

For some, enhancing employability and avoiding being "worse off in life" was more important than pursuing a degree that they were interested in and enjoyed.

"A lot of people are coming to uni now, including myself, for the idea that they think it will help them get a job in the future. It almost feels like if they don't take these routes they won't get as far as someone who did, so it's less about 'I came

here because I wanted to develop myself personally' and wanting to be better at their chosen subject because they enjoy it." Ppt 14

### 4.3. Social Support

This theme captured the importance of having a social support network. Irrespective of the source of support, a sense of connection, and knowing someone has got your back, were pertinent for QoL. However, *Family* and *Friendships and Peer Relations* were separate subthemes as many viewed them as distinct sources of support.

In the subtheme *Family*, participants discussed valuing their emotional and practical support, sometimes even more now they were at university, as they were (typically) living away from home for the first time. They were adjusting to maintaining relationships at distance and experiencing homesickness, and there was comfort in knowing that family would still be there even if they could not see them very often, a place of sanctuary away from university life.

> "I've always been really close with my family and I feel like having them—even though they're not here, I'll talk to them like nearly every night and I can Facetime them and just talking about what I've been doing and my life here and just knowing that I have that support system there even though they're not really here with me because when things do get too much and stressful and like I can just hop on a train and go home and have that break." Ppt 9

One participant who had not moved away for university referred to the "safety blanket" of having parents to go back to at the end of the university day, reiterating the sanctuary of family. At the same time, autonomy from family was part of their transition to independence.

> "With family I think it is important to keep in contact but also to let yourself be independent because it will be the first time for a lot of people that they have been away from their family for this long . . . we are not going to be living with our parents all our lives so it's good to get used to living away from them for a while." Ppt 15

The *Friendships and Peer Relations* subtheme captured feeling part of a wider support network with peers they were in close proximity to during university. Meaningful peer relationships were vital, particularly in the absence of close family. Having support from friends helped participants to navigate the new experiences, challenges and changeability of university life. Friends mitigated loneliness and offered a chance to take a break.

> "To have a good friendship circle [is most important for QoL]. I think it would be quite lonely without it, there are quite a lot of people at uni and it can be overwhelming . . . the course I'm doing is quite intense so you need somewhere you can just relax." Ppt 8

> "This place is constantly quite dynamic and quite changing so having friends around to help you feel better or give me something good to look forward to while I'm doing this kind of thing really makes me feel better when I'm doing it." Ppt 14

The dynamic and transient social contacts at university also made it difficult to form new connections and make friends.

> "I meet people and I know them for like two days and then they're gone and then there's new people and then I meet them and then they're gone. It's sort of an endless cycle and I don't know when it's going to stop." Ppt 1

### 4.4. Concerns about Finances and Financial Independence

These concerns were not just about having enough money to pay for essentials—although that was crucial—but having enough money to make the best of their time at university.

"Financial stresses, student loan, not covering bills or rent very well. It effects whether you can eat well, socialise well." Ppt 2

"If I do decide that I'm going to waste some money on going out that means there is a loss somewhere else so I can't buy as much food as I want to—I mean I'm not starving but it's just that sometimes if you want to go out somewhere and you have to sort of pick and choose what you want to do—but when you're completely independent it's sort of a new decision you have to make than when you're at home . . . it's quite a hurdle to live properly without someone telling you what to spend your money on." Ppt 15

Alongside concerns about learning how to budget and manage money, there was guilt and frustration about relying on parents. This made participants feel like they had less freedom over what to spend money on.

"I guess you are quite dependent on your parents and my parents are generous but we haven't really come to an agreement with money and things so it's quite up in the air because I feel bad asking for money because I do want to be independent but I often can't be." Ppt 3

Perceived differences in financial resources were a significant factor limiting one student's QoL compared to another's. Those who were better off were viewed as likely to be getting more out university life.

"If you have more finance behind you your quality of life and study will be better in a way you won't need to worry as much about finance." Ppt 16

### 4.5. Physical Environment

This theme was about the influence of where participants were living and studying. The first subtheme, *Clean, Safe and Social Living Space*, captured quality of accommodation, and the neighbourhood in which it was situated.

"Whether you live in a clean or dirty house, what your neighbourhood is like—I think I'm quite lucky in that respect as I have a decent house, live in a perfectly nice area near the centre of town which is very convenient." Ppt 3

Feeling safe was paramount, whether this was the security of the accommodation itself or the neighbourhood.

"I'm not that bothered about the state of where I live. As long as I feel safe in it." Ppt 10

The ease of getting to campus was important for QoL. Having poor transport links and/or living further away from university could be detrimental. The expense and time spent commuting made it trickier to stick around after taught classes to socialise and build networks.

"I'm probably not as heavily socially involved in uni as most people would be because a lot of the [clubs and societies] that would have interested me are really late at night. If you live just outside of campus that's fine, but I live at home which is like forty minutes to an hour drive so I'm not involved in them." Ppt 17

Accommodation was seen as somewhere to socialise (as well as study and relax), meaning that social dynamics with flat-mates and opportunities for social connection in accommodation were important.

"Sometimes being thrown in together and living with those people can cause a lot of clashes and can make you homesick and you know, some people might be more messy and you're not used to being in that kind of environment and that can cause clashes and generally have a negative effect." Ppt 9

"Our accommodation doesn't have a common room and other ones do and you can talk to people from outside your flat as well whereas we can only talk to each other because we don't really have a place to meet other people in the block." Ppt 15

The subtheme *Campus and the wider setting* firstly referred to the physical campus and its amenities and facilities, such as the accessibility and quality of study and teaching spaces.

"You spend most of your time at uni so if you don't necessarily enjoy the environment, be it the classroom next door is really noisy or people in the library being really noisy, that can obviously have an effect on you . . . it's not about being physically comfortable but psychologically as well. Being in a place where you feel like you could succeed." Ppt 18

This subtheme secondly included the university town/city, what it had to offer, and how safe it was.

"The city—it's very good for a student because everything is in walking distance . . . I think there is only one thing missing and this is the job opportunities for students." Ppt 13

"When considering this university I researched a lot about what [city] is famous for—what the surrounding area is like and the features and facilities to help support my degree. Crime rate I would look at—so if there is a low crime rate it's more appealing, I would try and study at a place with low crime rate because a high crime rate might affect a student's quality of life." Ppt 16

### 4.6. Physical and Mental Well-Being

Physical and mental well-being were separate subthemes as they were considered linked but distinct influences on QoL. *Having Sufficient Mental Well-being to Engage* across a continuum from stress to significant mental health difficulties was important for QoL across social and academic aspects of university.

"In relation to mental health it is strongly related [to QoL] because like, stress can cause a lot of things. And then affect academic grades and stuff." Ppt 6

"I've suffered from anxiety and depression for going on five years now, so I remember my first day walking into the lecture theatre [. . .] going from a class of 30 people to 500 . . . every day on campus is a bit of a project." Ppt 1

As well as directly influencing QoL by limiting opportunities, mental well-being could influence perceptions of QoL.

"Mental health affects quite a lot obviously your friends and family can be as loving as they want but if your mental health isn't sufficient it would affect your quality of life because you would deem everything to be not as—you wouldn't appreciate everything." Ppt 16

The subtheme *Being Healthy and Living Well* highlighted that good health enhanced QoL and meant that they could engage in university life.

"Health is important but you only notice it at our age if something goes wrong." Ppt 2

"When you are healthy you just accept it but when you don't have it you realise how much it affects you. It can affect your relationships or academic studies." Ppt 13

The recognition that health concerns detrimentally affect QoL by limiting opportunities was confirmed by those who had physical health problems.

"Substantively [influences QoL]? Like all the time. I didn't do as well in my exams because of my health." Ppt 1

Participants discussed the importance of a healthy lifestyle, maintaining fitness, eating well, and drinking less.

"Not just like good health or bad health but if you're at your optimum health and exercising and eating well it's easier to study. I think it is really important for your body obviously but also for your mind you need that stress relief that gives you those endorphins." Ppt 8

*4.7. Maintaining Balance*

Participants wanted to use the limited timeframe of their degree to make the best of the academic, social and personal development opportunities of university: "I'm here for uni but I don't want to just be here for uni."

Difficulties finding time to commit to all of the valued aspects of university life were captured by the subtheme *Time Management and Balancing Priorities.*

"As a student I'm always busy I feel like it's probably my personality as well as I'm always active but there is just so much to do. Like it's endless and I think that can have an impact on how much time you have with your friends and family, whether that's exercising, focusing on your health, extracurricular or studies so just more time." Ppt 8

"I want to do something extracurricular that looks good on my CV but with needing money and then also doing my course and needing a little bit of time to myself I feel like I just don't have that chance . . . I can't put all of my focus into my studies when I want to because I have to go off and have a part time job and I think that is probably the main thing that gets me down and with both of them together I don't have as much time to go home or see my family and that puts a lot of stress on because it makes me homesick as well." Ppt 9

Striking this balance was particularly challenging as participants had newfound independence concerning what they did and when.

"Twenties are the age in which you actually discover a lot of things for the first time and I mean there are things which you are meant to do in your twenties and you have to give time to them." Ppt 4

The subtheme *Opportunities to Take a Break* was about the importance of having the freedom to change pace and do something light-hearted and fun to reset and recharge their motivation.

"It's not about what you do but if it's during the week and it's a bad week, it's having something to look forward to and sometimes you just need to give your brain a break." Ppt 17

"It is important to get a few endorphins and get relaxed so that you can charge your energy and spend it on academic studies or whatever." Ppt 13

Participants found it difficult to have a (university) work/life balance and felt guilty about taking time off.

"I think I am quite good at my course, I'm doing ok at it but that's only because I'm dedicating so much time to it. If I give myself a weekend or some time off I know that I will start to fall back on the lead I've got. This means it's constantly got to be maintained and that is quite taxing. Just the knowledge that I've got to constantly keep working at a high level so I can maintain how good I am at it." Ppt 14

## 5. Discussion

The QoL dimensions in this study broadly overlap with existing QoL dimensions: health, social support and relationships, personal development and fulfilment, financial security, and the physical environment [9,10,12,18]. However, our conceptual model was

directly determined by students' perspectives and experiences. The unique liminality of being a student, emerging adults transitioning to independence, was evident across themes.

One of the findings that sets this model apart from prior QoL research is maintaining balance as a standalone dimension. While existing definitions emphasise the importance of a range of activities [9,10,19], here the need to balance these was a core factor influencing QoL. This might be because students are emerging adults managing competing demands independently for the first time, whereas working adults may be more used to this, and adolescents' lives are generally organised by adults.

Gaining independence—growing up, becoming more self-reliant, and managing transitions—was a specific facet of personal growth in our model. Students wanted to enjoy and do well in their studies, and academic pressures elicited a sense of purpose as well as self-doubt and stress. However, beyond their degree, students wanted to use their time at university to navigate the transition to independence and build skills and confidence for impending adulthood and to secure themselves the brightest future. This cut across themes, and there was conflict between becoming independent (e.g., living away from parents) and being dependent (e.g., need for emotional, practical and financial support from 'adults'). For example, students are concerned about managing money, but they are not fully independent as they do not have a significant personal income, like working adults, nor are they wholly dependent on family finances, like most adolescents.

Linked to this, existing measures assume that family are the people you live with, such as spouses and children among adults [12] and parents among adolescents [18]. Students are usually living away from family, with peers. This influenced the relative importance of sources of support. For those living away from home, social connection and support from peers was paramount. However, students also needed contact and encouragement from people older than themselves, as seen in adolescents [19,20]. Parents offered comfort and respite rather than day-to-day support, and university support, including relationships with lecturers ('adults'), was crucial.

Students are emerging adults, and physical health had a more preventative focus on keeping fit and healthy rather than the impact of existing symptoms. Mental well-being focused on their ability to engage with student life rather than symptoms. While tentative, the unique challenges to QoL could explain the poorer mental health of students compared to non-university attending peers before age 25 [21]. Student life is tumultuous, but graduates have entered the more 'stable' roles of adulthood, with fewer (or more manageable) demands, greater confidence, and less uncertainty. Supporting the specific challenges to QoL during the university journey is crucial to support student mental health and well-being during that time—and not only that, but helping them to get the best out of their university experience in the ways that are important to them.

## 6. Practical Implications and Future Directions

Although practical implications are preliminary due to the scale of the current study, our novel conceptual model of QoL aligns with the Whole University approach, and has pragmatic value for universities [2]. Firstly, the model brings together multiple, interconnecting factors relevant to student well-being in one model, highlighting the cross-cutting nature of the transition to independence. The transition into and across university, with its associated new social, academic, environmental and future-related stressors, is prominent in the University Mental Health Charter [2] and wider student well-being research [22–25].

Secondly, our model acknowledges the impact of significant mental and physical health problems while focusing on preventative, proactive support for well-being, including positive interactions with university staff. This highlights the need for better training for staff regarding pastoral support [23]. Students also turn to peers for support, and these friendships may carry greater importance and responsibility than at other stages of adulthood [23,26]. However, building support networks and stable, reliable bonds in the first place is tricky, as the move to university brings a dramatic increase in class size and

being moved around in small-group teaching [24,25]. Research on proactive peer support approaches is limited [27], and an important area for further exploration.

Our model identifies a number of further key areas where students would value support, reinforcing the wider literature as well as student priorities for mental health research [28]. Support with balancing competing demands is crucial. Students feel "squished" by demands while adjusting to university life, and experience uncertainty about how they are doing in relation to peers [22,24]. Managing financial concerns, the promotion of healthy behaviour, the living standards and opportunities for social connection available in accommodation, and opportunities for social connection more widely are important areas for universities to consider [22–24,29–31]. Improved provision in any of the domains would enhance university support, which itself was a facet of student QoL. As students indicated that dimensions of QoL were interconnected, improvements in one dimension could cascade into improvements in further dimensions.

Finally, by understanding what the ideals of student life are, students can assess where they view themselves relative to those ideals. Our model will inform the development of a theory-driven, co-produced measure to quantify the most prominent issues impacting student QoL, identify strengths and areas for improvement, and evaluate what works for enhancing student QoL. This should be done at scale to check the resonance of these QoL elements with a wider group of university students.

### 7. Limitations

While purposive sampling aimed to ensure that the samples reflected the student population at each university, participants were studying at a limited number of universities, which may have their own contextual factors, furthering the argument for wider-scale research. To focus on the developmental stage of emerging adulthood, participants were 18–25. This restricts the transferability of findings to mature students, who may experience different challenges yet have a stronger sense of independence.

### 8. Conclusions

This is the first multidimensional model of QoL that captures the context of being a university student. Our model highlights what is most important for students to feel that they are living a good life and getting the best out of university: a supportive university environment that allows them to manage academic pressures, do well, and enjoy their studies; transitioning to independence and opportunities to develop skills; making and having friends; maintaining connection with family; financial security; and decent physical places in which to live and learn. The scale of the current study makes support recommendations tentative but, combined with what we already know about student well-being, students are likely to benefit from activities (in and beyond the classroom) that help them to make social connections, and support with managing academic pressures, work–life balance, and the transitions into and across university life and beyond. These are universal initiatives with the potential to benefit all undergraduates, in line with the Whole University approach.

**Author Contributions:** Conceptualization, A.L.D. and N.B.; methodology, A.L.D., E.S. and N.B.; validation, A.L.D., N.B., E.S., J.M.A.M. and G.P.; formal analysis, A.L.D., N.B., G.P. and J.M.A.M.; investigation, A.L.D., N.B., J.M.A.M. and G.P.; data curation, A.L.D., N.B., J.M.A.M. and G.P.; writing—original draft preparation, A.L.D.; writing—review and editing, all authors; visualization, A.L.D.; supervision, A.L.D. and N.B.; project administration, A.L.D.; funding acquisition, A.L.D. and N.B. All authors have read and agreed to the published version of the manuscript.

**Funding:** This project was funded by a British Academy/Leverhulme Small Grant Award (SRG\170192). A.L.D. and N.B. were partially supported by ESRC funding (ES/S00324X/1).

**Institutional Review Board Statement:** The study was conducted in accordance with the Declaration of Helsinki, and approved by Psychiatry, Nursing and Midwifery Research Ethics Panel, King's College London (code LRS-17/18-7273).

**Informed Consent Statement:** Informed consent was obtained from all subjects involved in the study.

**Data Availability Statement:** Data from this study can be found at https://doi.org/10.6084/m9 .figshare.25434835.v1 (accessed on 27 March 2024).

**Acknowledgments:** The authors would like to thank Bridget Juniper (Work and Wellbeing Ltd.) and the Student Minds charity for paid consultancy in developing this study. We would also like to thank the students who took part.

**Conflicts of Interest:** The authors report there are no competing interests to declare.

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
