# Peer review of "How Can the University Environment Support Student Quality of Life? A Novel Conceptual Model"

_education, doi:10.3390/educsci14050547_

Round 1

Reviewer 1 Report

Comments and Suggestions for Authors

University is a period of change as young people develop new skills, experiences, expand social networks and gain knowledge. For many students going to university can be a stressful life event as they negotiate changes in lifestyle, community and relationships. The text concerns an important issue: the quality of life of university students. In recent years, the assessment of QoL became useful to determine the impact of illnesses/diseases and many interventions. However, little attention has been given to QoL of university students during their educational process, which is recognized as a high-stress period.

I evaluate the text positively, but the findings are not, in my opinion, groundbreaking, taking into account such a long tradition of research on the quality of life of students, or the issue of school/organizational climate.

The presented research should be treated as a preliminary study that requires continuation in the form of representative research in order to draw practical and far-reaching conclusions.

Author Response

Thank you for your positive appraisal of our research. While the findings resonate with prior work in the student mental health and well-being literature, to our knowledge this is the first to focus on conceptualising student quality of life specifically, from the perspective of students. However, we agree that this is a preliminary study, and have gone through our discussion to ensure that we are not making claims disproportionate to the scale and scope of our study. Our Conclusion already highlighted that “The scale of the current study makes support recommendations tentative”, but we made the following additions to reinforce this point earlier in the Discussion:

Lines 443-444: “Although practical implications are preliminary due to the scale of the current study, our novel conceptual model of QoL aligns with the Whole University approach, and has pragmatic value for universities.”

Lines 476-476: “This should be done at scale to check the resonance of these QoL elements with a wider group of university students.”

Lines 478-481: “While purposive sampling aimed to ensure the samples reflected the student population at each university, participants were studying at a limited number of universities, which may have their own contextual factors, furthering the argument for wider-scale research.”    

Reviewer 2 Report

Comments and Suggestions for Authors

Congratulations to the authors for a brilliant research towards building a novel conceptual model of student Quality of Life (QoL) specific to the emerging adulthood cohort attending university. Despite a small cohort of participants it was nevertheless a diverse sample in terms of gender, age, ethnicity, university location and student status. The methodological approach enabled in-depth exploration of diverse subject matter leading to a revelation of numerous themes and sub-themes. And the most significant aspect of the methodological approach was the application of participants knowledge in the construction of the Quality of Life (QoL)conceptual model. It was an empowering research process which generated knowledge directly relevant to the cohort's experiencing of university life. The findings of this research will make a valuable contribution towards understanding and attending to the needs of the emerging and adulthood cohort undertaking tertiary studies, in particular the aspect of maintaining balance between studies and other psycho-social necessities. A comparative study around those who fallout of tertiary studies in this cohort would be valuable in furthering understanding of this cohort's experiencing of University Quality of Life (QoL). One typo in sentence 331 please change the word 'effect' to 'affect'.

All the best

Author Response

Thank you for your positive review of our research and manuscript. We have fixed the typo in sentence 331 (changed effect to affect).